# Recent Advances in the Role of Osteocytes in Orthodontic Tooth Movement

**DOI:** 10.3390/ijms26199396

**Published:** 2025-09-26

**Authors:** Aseel Marahleh, Fumitoshi Ohori, Jinghan Ma, Ziqiu Fan, Angyi Lin, Kohei Narita, Kou Murakami, Hideki Kitaura

**Affiliations:** 1Division of Orthodontics and Dentofacial Orthopedics, Graduate School of Dentistry, Tohoku University, 4-1, Seiryo-Machi, Aoba-ku, Sendai 980-8575, Miyagi, Japan; fumitoshi.ohori.t3@dc.tohoku.ac.jp (F.O.); ma.jinghan.s1@dc.tohoku.ac.jp (J.M.); fan.ziqiu.q1@dc.tohoku.ac.jp (Z.F.); lin.angyi.r5@dc.tohoku.ac.jp (A.L.); kohei.narita.a2@tohoku.ac.jp (K.N.); kou.murakami.b2@tohoku.ac.jp (K.M.); 2Frontier Research Institute for Interdisciplinary Sciences, Tohoku University, Sendai 980-0845, Miyagi, Japan

**Keywords:** orthodontic tooth movement, osteocyte, osteoclast, bone resorption, cytokines, ageing, necroptosis, mechano-adaptation

## Abstract

Orthodontic tooth movement (OTM) is a biologically orchestrated process involving the dynamic interplay of mechanical force, inflammatory signaling, and bone remodeling. Osteocytes, the most abundant cells within the bone matrix, serve as mechanosensitive regulators that transduce mechanical cues into biochemical signals in response to orthodontic force. This review delineates the multifaceted role of osteocytes in facilitating bone resorption required for OTM. The role of osteocytes is examined in inflammation, mechanical adaptation, and cell death. Additionally, we discuss the evidence on how aging alters osteocyte function, with senescence-associated changes disrupting mechanosensory networks and attenuating bone remodeling. Finally, the possibility that osteocytes themselves undergo morphological adaptation during force application is explored. This structural plasticity may impact individual variability in orthodontic outcomes. Advancing our understanding of osteocyte signaling in OTM holds significant promise for optimizing treatment outcomes across diverse patient populations.

## 1. Introduction

Orthodontic tooth movement (OTM) is a dynamic process that involves complex interactions between mechanical force application, inflammatory responses within the periodontal ligament (PDL), and remodeling of the alveolar bone [1,2,3,4,5,6]. Central to this remodeling is the osteocyte, a highly specialized and mechanosensitive terminally differentiated cell embedded within the bone matrix [7,8]. As the most abundant cell type in bone, osteocytes comprise approximately 90% of the total bone cell population, and function as key transducers of mechanical and biochemical signals [9,10,11,12,13]. Osteocytes arise when mesenchymal stem cells derived osteoprogenitors differentiate into osteoblasts, a subset of which becomes embedded in the bone matrix and gradually matures into young then mature osteocytes [14]. This transition is marked by progressive morphological changes: osteoblasts flatten, extend dendritic processes, and integrate into the existing osteocytic network [15,16]. Once embedded, osteocytes occupy lacunae and extend numerous dendrites through canaliculi, forming the lacuno-canalicular system, an elaborate three-dimensional network that connects osteocytes with each other, as well as with bone surfaces, the bone marrow, and the vascular system [16,17]. This intricate architecture facilitates the diffusion of nutrients and oxygen and enables osteocytes to function as central regulators of mineral homeostasis and skeletal remodeling [10]. Depending on the type and magnitude of mechanical stimuli, osteocytes can promote either bone resorption or bone formation by modulating localized signaling to osteoclasts and osteoblasts [18,19,20,21,22,23,24]. In this capacity, they serve as integrators of mechanical and molecular cues, which are essential for the adaptive remodeling of bone in response to orthodontic forces. The next sections elaborate on the role of osteocytes in facilitating OTM with special emphasis on the mechanisms that lead to bone resorption on the compression side, discussing inflammatory signals, osteocyte death, and ageing, as well as external mechanical stimuli during OTM, emerging as a potential advantageous adjunctive to orthodontic load. Lastly, the overlooked osteocyte morphological shifting underload and in different locations is discussed.

## 2. Osteocyte Signaling Pathways Governing Osteoclastogenesis in OTM

Osteoclasts are the principal cells of bone resorption and play a critical role in the remodeling processes required for OTM [25]. These large, multinucleated cells arise from hematopoietic stem cells of the monocyte–macrophage lineage through a tightly regulated differentiation cascade. The process is initiated by the transcription factor PU.1, which drives the expression of c-fms, the receptor for macrophage colony-stimulating factor (M-CSF) [26]. M-CSF promotes the differentiation of monocytes into macrophages, which subsequently express RANK, the receptor for RANKL. RANKL binding to RANK is the essential step that commits osteoclast precursors to osteoclasts [27,28]. Both M-CSF and RANKL are indispensable for osteoclastogenesis, as demonstrated by severe osteopetrosis in models lacking either molecule or their receptors [18,29]. RANKL is secreted by osteoblasts, osteocytes, stromal cells, and activated T cells, and its activity is modulated by osteoprotegerin (OPG), a decoy receptor that prevents RANK activation [18,23,24,28].

Within this framework, osteocytes have emerged as key regulators of osteoclast-mediated bone resorption during OTM [30]. These mechanosensitive cells are ideally positioned to detect changes in mechanical loading and transduce those signals into biochemical responses [30]. Osteocyte-specific deletion of RANKL significantly reduces OTM distance and osteoclast numbers in vivo, highlighting their central role in directing bone resorption in response to compressive force [31]. Furthermore, osteocytes respond to inflammatory stimuli, particularly tumor necrosis factor-alpha (TNF-α), which is upregulated on the compression side during OTM [32,33,34,35,36,37,38]. TNF-α is a cytokine that is involved in osteoclastogenesis in synergy with M-CSF, under inflammatory conditions [39,40,41,42]. Mechanical loading during OTM stimulates local TNF-α release on the compression side. Mice lacking TNF-α receptors 1 and 2 showed reduced OTM distance compared to wild type, with receptor 2 acting as the primary transducer of TNF-α–driven bone resorption required for OTM [32,33]. TNF-α synergizes with M-CSF to promote osteoclastogenesis and has been shown to directly enhance osteocyte RANKL expression through activation of Extracellular signal-regulated kinases 1 and 2 (ERK1/2), c-Jun N-terminal kinase (JNK), p38 mitogen-activated protein kinase (p38 MAPK), and nuclear factor kappa-light-chain-enhancer of activated B cells (NF-κB) signaling [34]. Co-culture studies have shown that TNF-α-stimulated osteocytes promote osteoclast formation in a RANKL-dependent manner [34]. In vivo, mice deficient in TNF receptors exhibited impaired RANKL expression in osteocytes and reduced tooth movement, further confirming the direct effect of TNF-α on osteocyte RANKL expression and regulation of bone resorption [35].

TNF-α also amplifies osteoclastogenic signaling by increasing osteocyte expression of sclerostin, a Wnt antagonist known to suppress osteoblast differentiation while enhancing RANKL expression [37]. This dual action sustains a pro-resorptive microenvironment, positioning TNF-α as a central modulator of osteocyte-driven bone remodeling during OTM. Moreover, TNF-α increases RANK expression in osteoclast precursors, further sensitizing them to RANKL stimulation [43]. This interconnected network of signaling molecules of TNF-α-RANK-RANKL-sclerostin collectively defines a potent signaling cascade through which osteocytes mediate targeted resorption on the compression side. Recent data further elucidate this axis by demonstrating that mechanical compression upregulates *SOST* expression in alveolar osteocytes through paracrine cues derived from periodontal ligament (PDL) cells [44]. In vivo and in vitro findings reveal that PDL cells subjected to orthodontic force secrete factors that induce *SOST* upregulation in osteocytes without affecting their RANKL/OPG expression. Neutralization of sclerostin abolished this osteocytic response, confirming its mechanistic role in force-induced remodeling [44]. These findings underscore a spatially coordinated, cell-type-specific signaling loop between PDL cells and osteocytes, which converges on sclerostin as a key mediator of compressive bone adaptation, further compounding the osteoclastogenic potential of osteocytes through the RANK-RANKL-sclerostin axis.

Recent transcriptomic analysis has significantly deepened our understanding of how osteocytes respond to pro-inflammatory stimuli. RNA-seq of highly purified osteocytes from DMP1-Topaz mice exposed to TNF-α identified 422 differentially expressed genes, with 246 upregulated and 176 downregulated transcripts. Among the most upregulated genes were several pro-inflammatory mediators, including *Cxcl10*, *Ccl2*, *and Gpr84*, with *Cxcl10* ranking in the top ten. *Tnfsf11* (RANKL) was also upregulated, reinforcing the direct role of osteocytes in osteoclastogenesis under inflammatory conditions. In contrast, the downregulated genes included *Gbp5* (a GTPase involved in immune regulation), *Gjb4* (a connexin critical for intercellular communication), and both *Nos2* and *Clec4e*, which are associated with anti-inflammatory and homeostatic pathways. Together, these transcriptional shifts suggest that TNF-α drives a catabolic, pro-resorptive osteocyte phenotype by simultaneously enhancing osteoclastogenic signals and suppressing homeostatic regulators [45]. Mechanistically, TNF-α–stimulated osteocytes were shown to secrete CXCL10. Although CXCL10 did not directly promote osteoclast differentiation, it significantly enhanced the migration of osteoclast precursors through interaction with the CXCR3 receptor. Transwell migration and co-culture assays using TNF receptor-deficient precursors demonstrated that CXCL10 released from osteocytes is both necessary and sufficient to drive precursor chemotaxis. Moreover, in vivo administration of a CXCL10-neutralizing antibody attenuated TNF-α–induced osteoclast formation and bone resorption. Unlike RANKL, which is essential for osteoclast differentiation, CXCL10 functions primarily as a chemotactic factor to increase precursor availability. The study suggests that CXCL10 acts in concert with, rather than independently of, RANKL.

These findings position osteocytes as central mediators of inflammatory bone loss, not only through RANKL production but also by orchestrating osteoclast precursor recruitment via chemokine signaling. CXCL10 belongs to the CXC chemokine family, a group of small, secreted proteins characterized by two conserved cysteines separated by one amino acid (the “C-X-C” motif). CXC chemokines act as potent chemoattractants for immune cells such as monocytes and lymphocytes, functioning through G protein–coupled receptors like CXCR3. Specific ligand–receptor interactions, such as CXCL10–CXCR3, are increasingly recognized as critical regulators of inflammatory signaling and immune cell positioning [46] which are processes that are now known to extend to osteoclast precursor recruitment during OTM.

Additionally, regulated cell death in osteocytes has been identified as an additional mechanism influencing osteocyte osteoclastogenic signaling. Osteocyte apoptosis has been shown to facilitate osteoclastogenesis and alveolar bone resorption associated with OTM; however, blocking osteocyte apoptosis showed no effect on OTM distance [47]. In parallel, oxidative stress-induced apoptosis of osteocytes has been linked to bone resorption during OTM. Reactive oxygen species (ROS) and endoplasmic reticulum (ER) stress drive apoptosis in response to compressive force, and pharmacological inhibition of either ROS or ER stress significantly reduces osteocyte death and subsequent bone loss. However, studies using the apoptosis-inhibiting agent IG9402 found no significant impact on OTM, likely due to compensatory apoptosis in the periodontal ligament [48]. These findings suggest osteocyte apoptosis contributes to osteoclastogenesis; however, it may not be critical for tooth movement. Another form of regulated cell death, specifically, necroptosis, a programmed necrotic pathway characterized by receptor-interacting protein kinase 3 (RIP3) and mixed lineage kinase domain-like protein (MLKL) activation, has been shown to influence OTM [49]. This pathway has been implicated in driving osteoclastogenesis by osteocytes, as it has been shown that TNF-α induces necroptosis in osteocytes at compression sites, triggering the release of damage-associated molecular patterns (DAMPs). These DAMPs robustly promote osteoclast differentiation [50]. In TNFR1/2 knockout mice, inhibition of necroptosis resulted in reduced osteoclast numbers and diminished tooth movement, confirming its functional importance [49]. Thus, necroptotic signaling has emerged as an additional mechanism by which osteocyte death promotes osteoclastogenesis during OTM.

Moreover, systemic influences, such as sympathetic nervous system activity, also modulate osteocyte behaviour during OTM. In hypertensive rat models with elevated sympathetic tone, blocking β-adrenergic signaling during OTM reduced osteocyte expression of sclerostin and RANKL, decreased osteoclastogenesis on the compression side, and increased bone volume in the bifurcation area of the maxillary molars [51]. This supports a regulatory role for neural input in osteocyte-mediated control of osteoclast formation and bone resorption during OTM.

Lastly, mechanical tension induces autophagy in osteocytes as an adaptive response that enhances RANKL secretion and supports bone resorption under compressive force [51]. Conversely, tensive stress also activates autophagy in osteocytes and drives osteogenesis via fibroblast growth factor 23 (FGF23) [52]. Under compressive stress, autophagy is activated in osteocytes via suppression of Rho-associated coiled-coil containing protein kinase (ROCK) and reduced AKT phosphorylation, leading to nuclear translocation of transcription factor e3 (TFE3), a transcription factor that upregulates autophagy-related genes such as autophagy-related 7 (ATG7) and microtubule-associated protein 1 light chain 3 beta (LC3B). In vivo models confirmed that interfering with autophagy impairs OTM, placing this pathway alongside apoptosis and necroptosis in the osteocyte’s mechanoresponsive toolkit [53]. In summary, in vivo OTM studies and in vitro studies reveal that osteocytes integrate mechanical, local inflammatory, and systemic cues through interconnected signaling pathways to orchestrate precise site-specific osteoclastogenesis during OTM (Table 1).

## 3. Osteocyte Mechanotransduction and Mechanical Modulation of Bone Resorption in OTM

Several mechanical stimuli have been identified as critical modulators of osteocyte behavior during OTM. Fluid-flow shear stress, generated by strain-induced interstitial fluid movement within the lacuno-canalicular system, activates osteocyte mechanotransduction pathways, including ERK1/2 and Wnt/β-catenin, promoting RANKL expression and osteoclastogenesis [54,55]. Compressive strain and hydrostatic pressure resulting from orthodontic force lead to localized osteocyte apoptosis, particularly on the compression side, which further recruits osteoclasts via damage-associated signaling [54]. Additionally, direct matrix deformation alters the cytoskeletal structure and integrin signaling in osteocytes, influencing their secretion of RANKL, sclerostin, and other remodeling factors [54]. Notably, the mechanosensitive ion channel PIEZO1 was once regarded as a key regulator, as its activation by membrane stretch induces RANKL expression [55,56] however, later reports showed that its effect may be redundant, as explained below [57].

Most recently, low-magnitude high-frequency vibration (LMHFV) has emerged as a potent adjunctive mechanical signal capable of accelerating OTM, primarily by modulating osteocyte behaviour on the compression side of the alveolar bone [58]. Experimental studies have demonstrated that LMHFV enhances osteoclastogenesis through osteocyte-mediated signaling. Sasaki et al. reported that LMHFV combined with static orthodontic force significantly accelerated tooth movement in rats, an effect abolished by inhibition of the TGF-β receptor, identifying osteocyte-derived TGF-β1 as a key mediator [59]. TGF-β1 plays a dual role in osteoclastogenesis, depending on concentration, timing, and cellular context [60,61]. At low levels, it supports early osteoclast differentiation by upregulating RANK expression and promotes maturation and resorptive function by increasing expression of nuclear factor of activated T cells (Nfatc1), tartrate-resistant acid phosphatase (Trap), cathepsin K (Ctsk), and dendrocyte-expressed transmembrane protein (Dc-stamp). Simultaneously, it stimulates osteoblast-coupling factors such as Wnt1, Wnt10b, and Sphk1, suggesting it contributes to balanced remodeling [62]. However, at higher concentrations or later stages, TGF-β1 can suppress osteoclastogenesis by inducing osteoclast apoptosis and increasing OPG expression while suppressing *SOST*, thereby decreasing the RANKL/OPG ratio [61]. Mechanistic in vitro data using MLO-Y4 osteocytes confirmed that LMHFV upregulates TGF-β1 and activates the NF-κB signaling pathway, leading to increased RANKL expression [58]. Co-culture of vibrated osteocytes with RAW264.7 pre-osteoclasts significantly enhanced osteoclast formation, an effect abolished by TGF-β1 inhibition [59]. In vivo, NF-κB–activated, RANKL-positive osteocytes were localized to the compression side during LMHFV-assisted OTM, emphasizing the spatial specificity of osteocyte mechanotransduction in response to LMHFV [59]. These findings collectively demonstrate that vibration enhances bone resorption in OTM via site-specific, TGF-β1 and NF-κB mediated osteocyte signaling. While TGF-β1 and NF-κB signaling pathways underscore the osteocyte’s responsiveness to vibration, the role of mechanosensitive ion channels in this process remains nuanced. Although Piezo1 is expressed in osteocytes and known to mediate mechanical signaling in other contexts [12,56,63,64,65], its deletion in Dentin matrix protein 1-Cre (Dmp1-Cre) conditional knockout mice did not impair tooth movement or periodontal ligament remodeling during OTM. Despite showing increased basal osteoclast numbers and alveolar bone loss, Piezo1-deficient mice exhibited normal responses to orthodontic force [57]. These results challenge the presumed centrality of Piezo1 in osteocyte mechanotransduction during OTM and point toward alternative or redundant signaling pathways. Another proposed mechanosensory structure in osteocytes is the primary cilium, a solitary, non-motile projection known to detect fluid shear stress and regulate expression of mechanosensitive genes such as prostaglandin E2 (PGE2), cyclooxygenase 2 (COX-2), osteopontin, and RANKL [66]. While in vitro studies suggest that elongation of cilia enhances osteocyte responsiveness to mechanical stimuli [66], in vivo data tell a different story. Osteocyte-specific deletion of intraflagellar transport protein 80 (*ift80*), a critical intra-flagellar transport protein required for ciliogenesis, did not impair OTM or alter bone remodeling parameters, including sclerostin and RANKL expression. This raises the possibility that primary cilia may not function as indispensable mechanosensors in the mineralized matrix, potentially due to spatial constraints or compensatory signaling mechanisms within osteocytes and surrounding cells. In conclusion, osteocyte-mediated mechanotransduction during OTM involves complex, context-dependent signaling cascades, with TGF-β1, NF-κB, and RANKL emerging as key effectors, while Piezo1 and primary cilia appear to play dispensable or redundant roles in vivo.

## 4. The Impact of Aging Osteocytes on OTM

OTM is fundamentally driven by an inflammatory response in the PDL and alveolar bone, which orchestrates the tightly balanced bone resorption and formation. However, this balance becomes increasingly dysregulated with age [67]. Aging introduces a series of molecular, cellular, and structural alterations that diminish the efficiency of bone remodeling and have been shown to slow the biological response to orthodontic forces [67,68,69,70,71]. A key hallmark of aging is cellular senescence, a state of irreversible cell cycle arrest triggered by various stressors, including telomere shortening, DNA damage, oxidative stress, oncogenic signaling, and mitochondrial dysfunction [72,73]. Unlike quiescent or terminally differentiated cells, senescent cells remain metabolically active but exhibit major shifts in gene expression, chromatin structure, and intercellular signaling [73]. They secrete a senescence-associated secretory phenotype (SASP), composed of pro-inflammatory cytokines, growth factors, and matrix-degrading enzymes that can profoundly affect surrounding tissues [73]. Osteocytes, as long-lived and deeply embedded bone cells, are particularly vulnerable to age-related senescence [74,75,76]. Irradiation-induced models have revealed that senescent osteocytes display classic features such as senescence-associated β-galactosidase (SA-β-Gal) activity, upregulated p16INK4a and p21CIP1, and the SASP profile, marked by significant transcriptional upregulation of proinflammatory and matrix-remodeling factors. In osteocytes isolated directly from aged mice, 23 out of 36 established SASP genes were significantly upregulated, including *Il6*, *Il8*, *TNF*, *Ccl2* (encoding MCP-1), *Ccl5* (encoding RANTES), *Csf1*, *Csf2*, *Csf3*, *Hmgb1*, *Serpine1* (encoding PAI-1), and multiple matrix metalloproteinases (MMPs). These factors are regulated in part by elevated NF-κB and IL-1α signaling, consistent with known SASP activation signals. Additionally, senescent osteocytes exhibited hallmarks of genomic instability, including telomere dysfunction-induced foci (TIFs) and senescence-associated distension of satellites (SADS). Importantly, markers of autophagy such as autophagy-related gene 7 (*Atg7*) and Microtubule-associated protein 1 light chain-3-alpha (*Map1lc3a-LC3*) were significantly suppressed in osteocytes with age, supporting a mechanistic link between impaired autophagy and SASP activation [75].

Beyond these hallmarks, recent studies demonstrate that senescent osteocytes undergo profound morphological and mechanotransductive alterations, including cytoskeletal stiffening, expansion of F-actin areas, and dendritic decline, impairing force sensing and ECM interactions [77]. Proteomic profiling of human bone further confirms that aging osteocytes activate inflammatory pathways while repressing osteogenic and metabolic programs, with rapamycin shown to partially reverse senescence phenotypes in vitro [78]. In young animals, senescent osteocytes initiate bone resorption by recruiting osteoclast precursors through secreted factors such as IGF1, MMP2, and MCP-1 [79]. Mechanistically, senescent osteocytes expressed elevated levels of SA-β-Gal and upregulated canonical senescence markers, including p16INK4a and p21CIP1, and transcriptomic profiling revealed enrichment of matrix-remodeling enzymes (MMP2), growth factors (IGF1), and chemotactic cytokines (MCP-1/CCL2). Functionally, conditioned media from podoplanin-positive senescent osteocytes promoted directed migration of osteoclast precursors in transwell assays, and co-culture experiments confirmed their differentiation into osteoclasts [79]. These findings point to a paracrine signaling axis in which senescent osteocytes establish a localized microenvironment favourable for resorption. Importantly, this early senescence-driven secretory response mirrors features of SASP observed in aging, yet in the young context, it may be transient and tightly regulated, acting in initiating the resorption phase of bone remodeling. This positions osteocyte senescence not solely as a pathological hallmark of aging, but as a context-dependent mechanism in initiating bone resorption required for bone remodeling [79].

Senescent osteocytes exhibit a reduction in the number of dendritic processes, which compromises the osteocyte network connectivity and mechanotransduction efficiency [80]. Senescent osteocytes are more spherical, small, and accompanied by cytoplasmic lysis, empty cytoplasm, and nuclear abnormalities [81]. These changes coincide with a 30% decrease in lacunar volume and a marked decline in lacunar density, suggesting widespread osteocyte loss in senescent tissues [82,83,84]. Moreover, aging bones show hypermineralized occluded lacunae, a process known as micropetrosis, indicating osteocyte death and mineral infill of empty lacunae [83,85]. These structural changes, together with disrupted perilacunar remodeling, reduce osteocyte sensitivity to mechanical stimuli, impairing their ability to detect orthodontic forces and regulate remodeling responses. Although direct studies of osteocyte aging in OTM are lacking, clinical and experimental data strongly support age-associated impairment in tooth movement. Adults exhibit slower rates of OTM compared to adolescents despite showing an elevated inflammatory profile, including TNF-α and osteoclast-related markers, including RANKL and matrix MMPs, indicating that the downstream bone remodeling machinery is blunted [75]. Adult patients have been reported to have reduced OTM response alongside increased discomfort, suggesting a disconnect between inflammation and effective remodeling [86]. It has been shown that aged rats showed reduced expression of autophagy-related proteins and elevated senescence markers. Pharmacological activation of autophagy with rapamycin restored osteoblast and osteoclast activity, enhanced osteogenic markers, and significantly improved OTM efficiency, highlighting the interplay between senescence, autophagy, and osteoblast-osteoclast interaction in OTM [87]. Aging also induces broader impairments: increased senescence in PDL cells, decreased vascularity, lower regenerative capacity, and a hyperinflammatory environment. Alveolar bone in aged animals demonstrates greater mineral density but reduced responsiveness to mechanical stress, linked to impaired activation of osteoclasts and osteoblasts. Additionally, osteoclastogenesis in aged animals is less responsive to RANKL and TNF-α stimulation, further compounding the remodeling deficit [71]. Despite substantial work on osteoblasts, osteoclasts, and PDL cells, the contribution of osteocyte aging to these phenomena remains underexplored. Given their central role as mechanosensors and regulators of both bone formation and resorption, it is highly plausible that dysfunctional osteocytes contribute significantly to the diminished responsiveness seen in older individuals undergoing orthodontic treatment. The discussed evidence suggests that aging reduces osteocyte viability, dendritic connectivity, and mechano-transduction. These deficits may impair force perception and downstream coordination of bone remodeling during OTM. However, direct experimental evidence on how osteocyte senescence leads to decreased resorption and OTM is required (Figure 1).

## 5. Osteocyte Morphology and Mechano-Adaptation During OTM

Osteocytes’ morphology reflects both genetic programming and environmental demand [88,89,90,91]. During OTM, the alveolar bone is subjected to mechanical forces that stimulate continuous remodeling. Osteocytes transduce these forces into biochemical cues that regulate localized bone resorption and formation. Yet, the question remains: do osteocytes themselves structurally adapt in response to orthodontic force? Transcriptomic analyses have identified a conserved osteocyte gene signature across multiple skeletal sites, suggesting a core identity linked to different patterning gene programs [88]. Differential expression of patterning and developmental genes also indicates that osteocyte phenotype is fine-tuned by local mechanical and structural contexts. Morphologically, osteocytes exhibit remarkable plasticity. In woven bone, they tend to be spherical and randomly oriented; in lamellar bone, they elongate and align with the principal loading direction [89,92]. Aging further alters this landscape; osteocytes become more spindle-shaped, exhibit diminished dendritic complexity, and are present at lower densities [90,91]. Interestingly, these changes correlate more with the age of the surrounding bone matrix than with the chronological age of the organism, suggesting that long-term mechanical strain shapes osteocyte morphology [93]. Genetic evidence reinforces the intimate link between osteocyte structure and function. Disruption of genes such as Osterix (*Sp7*), *Dmp1*, *Klotho*, Von Hippel–Lindau tumor suppressor (*VHL*), and *Sirt3* (Sirtuin 3) leads to impaired dendrite formation, distorted lacunar architecture, and defective mechanotransduction, resulting in dysregulated bone remodeling [94,95,96,97,98]. Regional morphological variations, such as early-onset lacunar mineralization in the auditory ossicles, likely represent functional adaptations to site-specific mechanical strain and the need or lack thereof for bone remodeling around osteocytes where mechanical load is lacking [99]. Pathological conditions also reshape osteocyte morphology. In osteoarthritis, osteoporosis, and osteopenia, high-resolution Three-dimensional nanoscale computed tomography (3D nano-CT) imaging has revealed significant alterations in osteocyte size, volume, shape, and dendritic connectivity [100]. These deviations reflect not only altered strain environments but also impaired mechanosensory function. Similarly, the blunted bone formation response to orthodontic force in aged bone may be partially explained by such structural remodeling of osteocytes. What remains poorly understood is whether osteocyte morphology dynamically adapts during OTM. While osteocyte involvement in OTM-specific bone remodeling is well established, the possibility that their own shape, orientation, and dendritic architecture shift in response to applied force remains underexplored. Preliminary evidence addressed this gap using a combination of finite element modeling and histomorphometric analysis, in a study demonstrating that osteocytes in regions subjected to unidirectional tensile strain in the jawbone exhibited increased surface area and cranial–caudal alignment [101]. Although overall density and elongation remained unchanged, the alignment and surface area shifts paralleled the local strain profile, strongly suggesting functional morphological adaptation [101]. These findings lend support to the hypothesis that osteocyte morphology is not merely passive or descriptive, but dynamically responsive to force, potentially influencing the efficiency of mechanotransduction. In the context of OTM, where force vectors are carefully applied and monitored, such morphological plasticity could modulate site-specific remodeling efficiency and explain interindividual variability in treatment outcomes, particularly in aged or diseased bone. In conclusion, emerging evidence suggests that osteocyte morphology is a functional variable shaped by the mechanical environment, and its adaptation during OTM may be a previously underappreciated modulator of bone remodeling dynamics.

## 6. Conclusions, Limitations, and Future Directions

The collective evidence discussed in this review underscores the new advances central to the role of osteocytes in orchestrating OTM (Figure 2 and Table 2). Osteocytes function as both mechanosensors and signaling hubs, integrating mechanical, inflammatory, and age-related cues to regulate bone remodeling with spatial and temporal precision. Recent transcriptomic and mechanistic data stress on the role of inflammatory mediators such as TNF-α and downstream molecules like RANKL and OPG, through which osteocytes tightly control osteoclastogenesis, and signals such as sclerostin and FGF23, which modulate bone formation during OTM. Apart from the various mechanical modes that affect osteocyte signaling, exogenous mechanical stimuli, LMHFV, have been shown to directly modulate and activate parallel mechanotransduction pathways in osteocytes, namely, TGF-β1, which further control bone resorption and remodeling, these data are promising and need to be verified in the clinic.

Cell death namely, necroptotic death of osteocytes, via RIP3–MLKL activation and DAMP release, represents an alternative osteoclastogenic mechanism, expanding the functional repertoire of these cells in response to stress. We also discussed the overlooked potential of shifting osteocyte morphology, which emerged as a potential determinant of mechanosensitivity, with variations in shape, dendritic complexity, and alignment likely influencing signal output. However, direct evidence linking osteocyte morphological adaptation to orthodontic force application remains lacking. Similarly, while age-related impairments in PDL and osteoclast activity are well documented, the contribution of osteocyte aging to delayed or altered OTM has not been thoroughly investigated. Despite compelling mechanistic insights, several limitations constrain the direct clinical translation of current findings on osteocyte-mediated OTM. The majority of studies are conducted in rodent models, which may not fully recapitulate the complexity and mechanical environment of alveolar bone loading in humans, thereby limiting extrapolation to clinical orthodontics. The beneficial effects of vibration therapy (LMHFV) on accelerating OTM are promising and mechanistically clear, but the results are largely based on small animal studies and short observation periods. Optimal parameters, safety, and patient compliance in clinical settings remain poorly defined. The role of osteocyte senescence and the SASP phenotype in osteocyte-driven resorption remains incompletely characterized. The extent to which senescence contributes to clinical OTM, particularly in adult or aging patients, is poorly understood, and its causality to blunted orthodontic response is unclear and may have profound consequences on root resorption seen in these patients, particularly with the advent of techniques that promise to accelerate tooth movement [102]. This mode of osteocyte ageing should be studied and awaits clinical validation. Additionally, there is a lack of osteocyte-targeted interventions in aging models that could confirm whether restoring osteocyte function can meaningfully improve OTM outcomes in older populations.

Emerging data on osteocyte senescence provide avenues for translational intervention in orthodontics. Senolytic agents such as dasatinib plus quercetin and fisetin selectively clear senescent osteocytes, improving bone quality and suppressing pathological bone loss in aging, diabetes, and cancer metastasis models [76,103,104]. Senomorphic strategies, including rapamycin and NF-κB inhibitors, attenuate SASP while preserving cell viability, restoring osteoblast/osteoclast balance, and improving mechanotransduction [71]. Targeted manipulation of osteocyte senescence during OTM could offer novel clinical benefits.

Outstanding Questions and Future Directions

Should osteocyte-targeted interventions in aging models, particularly the use of senolytics, be pursued despite significant safety concerns such as off-target toxicity and the heterogeneity of osteocyte populations?Rigorous translational and clinical studies are needed to evaluate the efficacy and safety of adjunctive mechanical stimulation strategies, such as low-magnitude high-frequency vibration (LMHFV), in orthodontic patients.At a fundamental level, the field lacks advanced single-cell and spatial omics approaches capable of mapping osteocyte heterogeneity within bone. Applying these tools to human alveolar bone under varying mechanical strain, and in aged or diseased contexts, would provide a nuanced understanding of osteocyte biology. Such insights are critical to develop personalized, mechanism-based orthodontic therapies.

## Figures and Tables

**Figure 1 ijms-26-09396-f001:**
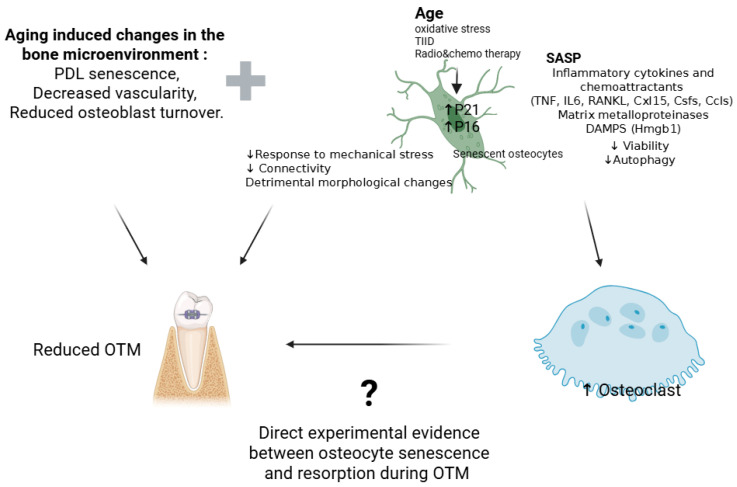
Schematic diagram exploring the mechanisms by which aging and osteocyte senescence contribute to osteoclast formation and OTM. Direct mechanistic evidence on the role of osteocyte senescence in osteoclast formation and bone resorption during OTM is lacking, which is denoted by the question mark.

**Figure 2 ijms-26-09396-f002:**
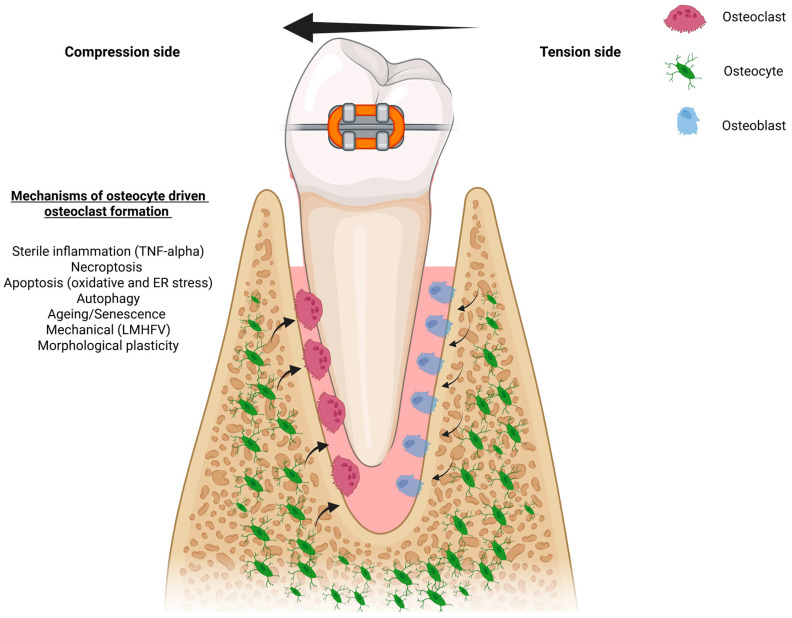
Schematic diagram of applied orthodontic force with osteocyte-activated pathways on the compression side, where osteoclasts resorb the bone and osteoblasts lay bone on the tension side.

**Table 1 ijms-26-09396-t001:** Summary of evidence derived from in vivo OTM intervention vs. general bone metabolism studies.

Pathway/ Factor	Evidence in In Vivo OTM Models	Evidence from Broader In Vivo/In Vitro Studies
RANK–RANKL–OPG	Osteocyte-specific RANKL deletion reduces OTM distance [31]. RANKL expression upregulated in osteocytes at compression sites during OTM [35]. β-adrenergic blockade reduced osteocyte RANKL expression and osteoclastogenesis on the compression side [50].	Osteocyte RANKL is essential for bone remodeling [18] Osteocyte-specific RANKL knockout mice develop osteopetrosis at 12 weeks of age [24].
TNF-α	TNFR1/2 KO have reduced osteocytic RANKL and sclerostin expression on the compression side [35,37].	TNF-α directly enhances osteocyte RANKL and sclerostin expression via MAPKs and NFκB pathways [34,37].
Sclerostin (*SOST*)	OTM induces sclerostin expression in osteocytes on the compression side [37,44]. β-adrenergic blockade reduced sclerostin expression on the compression side and increased bone volume [50].	Sclerostin increases RANKL expression [37] and neutralizing sclerostin increases OPG expression and decreases RANKL/OPG ratio [44].

**Table 2 ijms-26-09396-t002:** Summary of the evidence discussed on osteocyte-activated pathways in response to orthodontic force.

Stimulus	Intracellular Pathway/Process	Outcome	**Source**
**Mechanical compressive force and Inflammation**	Necroptosis via RIP3/MLKL activation	DAMPs release → RANKL-independent osteoclast differentiation	[49]
**Mechanical compressive force**	Autophagy induction (ROCK↓ → AKT↓ → TFE3 → ATG7 & LC3B upregulation)	↑ RANKL secretion → enhanced osteoclastogenesis & localized bone resorption	[52]
**Inflammation (TNF-α, IL1β)**	MAPK (ERK1/2, JNK, p38) & NF-κB activation	↑ RANKL & ↑ sclerostin → synergistic boost in osteoclastogenesis	[35,37]
**Exogenous mechanical stimuli (LMHFV)**	TGF-β1 release & NF-κB signaling	↑ RANKL expression → accelerated osteoclast differentiation during vibration-assisted OTM	[58,59]
**Sympathetic (β-adrenergic) stimulation**	β-adrenergic signaling → upregulation of sclerostin and RANKL	↑ Sclerostin & RANKL → shifts remodeling toward resorption over formation	[51]
**Aging (osteocyte senescence)**	SASP (p16INK4a/p21CIP1-mediated growth arrest) producing IL-6, IL-8, RANKL, etc.	SASP cytokines + RANKL → recruit & promote osteoclast precursors → age-related bone loss	[75,76]

## Data Availability

Data are available from the corresponding authors upon reasonable request.

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
