# Peer review of "Recent Advances in the Role of Osteocytes in Orthodontic Tooth Movement"

_ijms, 2025, doi:10.3390/ijms26199396_

Round 1
Reviewer 1 Report
Comments and Suggestions for Authors
The review is comprehensive and covers a wide spectrum of osteocyte biology in orthodontic tooth movement (OTM).
Comments:
- Abstract: The abstract is well written but somewhat lengthy. It may be improved by focusing more on the novel aspects of the review, such as necroptosis, osteocyte mechano-adaptation, and aging-related changes.
- Signaling Pathways (Section 2): While the role of TNF-α and RANKL is presented in detail, it would be helpful to more clearly distinguish between findings obtained from in vivo orthodontic models and those from broader osteoimmunology studies. In addition, the importance of osteocyte-derived CXCL10 in chemokine-mediated recruitment compared with RANKL signaling for osteoclastogenesis could be clarified. Are there comparative studies addressing this point?
- Cell Death Pathways: The discussion of apoptosis and necroptosis is stimulating, but it remains uncertain whether necroptosis functions entirely independently of RANKL in vivo, or whether there is a synergistic interaction. Further clarification on this issue would strengthen the section. Moreover, what is known about the relative contribution of apoptosis versus necroptosis to the overall process of OTM?
- Aging (Section 4): The connection between osteocyte senescence and reduced efficiency of OTM is convincing. However, most of the cited data are derived from general bone aging models. Have any direct orthodontic models been used to demonstrate senescence-related changes in osteocytes? Finally, what might these findings mean for clinical orthodontics in older patients—should treatment approaches or adjunctive strategies be reconsidered?
Author Response
Reviewer 1
Thank you for reading carefully and for the very helpful comments. I will answer them in order.
Comments:
Abstract: The abstract is well written but somewhat lengthy. It may be improved by focusing more on the novel aspects of the review, such as necroptosis, osteocyte mechano-adaptation, and aging-related changes.
ïƒ We shortened the abstract to focus on the novel aspects as recommended.
Signaling Pathways (Section 2): While the role of TNF-α and RANKL is presented in detail, it would be helpful to more clearly distinguish between findings obtained from in vivo orthodontic models and those from broader osteoimmunology studies. In addition, the importance of osteocyte-derived CXCL10 in chemokine-mediated recruitment compared with RANKL signaling for osteoclastogenesis could be clarified. Are there comparative studies addressing this point?
ïƒ Point1: To address the first point, we added table 1 which distinguishes between findings obtained in OTM models and findings obtained from in vivo or in vitro model that were not directly assessed by OTM.
ïƒ Point 2: To our knowledge, direct comparative studies quantifying the relative contributions of RANKL versus CXCL10 in osteocyte-driven osteoclastogenesis in OTM are lacking. Osteocyte-derived RANKL is indispensable for osteoclast differentiation as demonstrated in multiple in vivo models that we discussed. In contrast, CXCL10 appears to play a complementary role by recruiting osteoclast precursors to resorptive sites. Unlike RANKL, which directly commits precursors to the osteoclast lineage, CXCL10 enhances precursor availability through chemotaxis without inducing differentiation. Neutralization of CXCL10 reduced osteoclastogenesis [#45 in the review], but the effect was less pronounced than RANKL deletion in vitro, underscoring that CXCL10 elicits a milder response and acts in concert with, rather than independently of, RANKL. A direct comparative study in OTM models is still needed to define the relative weight of these two pathways.
We have added the following text line 141-144: The study suggests that CXCL10 acts in concert with, rather than independently of, RANKL. Unlike RANKL, which is essential for osteoclast differentiation, CXCL10 functions primarily as a chemotactic factor to increase precursor availability.
Cell Death Pathways: The discussion of apoptosis and necroptosis is stimulating, but it remains uncertain whether necroptosis functions entirely independently of RANKL in vivo, or whether there is a synergistic interaction. Further clarification on this issue would strengthen the section. Moreover, what is known about the relative contribution of apoptosis versus necroptosis to the overall process of OTM?
ïƒ Point 1: We agree with the reviewer. In actuality, multiple factors; including TNF-α, RANKL, sclerostin, and various forms of osteocyte death, synergize in regulating osteoclastogenesis during OTM. To avoid potential confusion, we have revised the section by removing the statement suggesting independence and replacing it with conclusions that more accurately reflect the current evidence.
ïƒ Point 2: Many studies have elucidated the role of osteocyte apoptosis in OTM, while the recent study by Ohori et al. (2025) specifically addressed osteocyte necroptosis in osteocyte-driven osteoclastogenesis and bone resorption during OTM (Ohori, SciRep, 2025). However, because cell death in vivo rarely occurs through a single pathway, but rather through intertwined mechanisms such as apoptosis and necroptosis, their relative contributions are difficult to quantify. We reckon that definitive mechanistic studies that selectively inhibit apoptotic versus necroptotic signaling in osteocytes are technically challenging in an in vivo OTM model. For a broader overview, we would like to refer the reviewer to a recent review discussing osteocyte death and its link to osteoclastogenesis, although not specifically within the realm of OTM(PMID: 40165960).
Aging (Section 4): The connection between osteocyte senescence and reduced efficiency of OTM is convincing. However, most of the cited data are derived from general bone aging models. Have any direct orthodontic models been used to demonstrate senescence-related changes in osteocytes? Finally, what might these findings mean for clinical orthodontics in older patients—should treatment approaches or adjunctive strategies be reconsidered?
- To our knowledge as of the writing of this review, no osteocyte senescence specific studies in OTM models have been conducted, although it is a very important outstanding question to tackle. In the discussion we do touch briefly on the potential role of senolytics as an adjunct treatment in older patients however, efficacy and safety studies remain a barrier to implementing them in clinical practice.
Reviewer 2 Report
Comments and Suggestions for Authors
1.How do the authors reconcile the discrepancies between in vitro mechanistic studies and in vivo findings, especially concerning the role of mechanosensors like Piezo1 and primary cilia?
2.Could the authors expand on potential osteocyte-targeted interventions in aging models and outline future application?
3.Please added more figures to illustrated the findings.
Author Response
Reviewer 2
Thank you for reading carefully and for the very helpful comments. I will answer them in order.
Comments and Suggestions for Authors.
1.How do the authors reconcile the discrepancies between in vitro mechanistic studies and in vivo findings, especially concerning the role of mechanosensors like Piezo1 and primary cilia?
ïƒ In vitro data discuss mechanistic pathways in isolation, for example, stretching osteocyte membrane activates RANKL signaling which if we take this logic further means that this may have an effect in vivo. But as we discuss a lot of this evidence in the review, multiple pathways converge on releasing factors that contribute to the cytokine milieu that is required to recruit osteoclasts for bone resorption. In the review section on mechanotransduction we describe that Piezo1 and cilia activation as redundant pathways as they are not the only nor the primary pathways that contribute to bone resorption during OTM, these conclusions are grounded in the evidence cited in our review.
2.Could the authors expand on potential osteocyte-targeted interventions in aging models and outline future application?
ïƒ We have touched on this subject discussing the potential role of senolytics as an option in OTM treatment in the older population, however efficacy and safety studies remain a barrier.
3.Please added more figures to illustrated the findings.
ïƒ We have added tables and figures in specific sections of the review.
Reviewer 3 Report
Comments and Suggestions for Authors
1. While the review is comprehensive, it seems overly focused on TNF-α signaling and necroptosis, reflecting primarily the authors’ own recent publications. Other pathways (e.g., prostaglandins, nitric oxide, Wnt signaling) receive less attention, which skews balance.
2. Some sections (e.g., on osteocyte morphology and mechano-adaptation, Section 5) are speculative with limited experimental evidence cited, yet the tone suggests strong conclusions.
3. Multiple sections repeat the role of RANKL/OPG and TNF-α signaling (Sections 2 and 3, again in Section 6 conclusions). The narrative could be tightened by merging overlapping parts.
4. The review highlights necrosis/necroptosis as an “alternative RANKL-independent mechanism” (lines 164–174) but does not adequately discuss limitations—most data are from mouse models, and human validation is absent.
5. The LMHFV (low-magnitude high-frequency vibration) section presents vibration therapy as promising (lines 209–231), but the authors themselves admit limited evidence (rodents, short-term). Still, the conclusions read stronger than warranted.
6. Aging effects are presented as largely due to osteocyte senescence (lines 267–317), but alternative explanations (vascularity loss, systemic hormonal changes, reduced PDL turnover) are only briefly acknowledged.
7. Table 1 and Figure 1 both appear only at the very end (in the Conclusions, Limitations, and Future Directions section). This is unusual for a review article, where figures and tables are normally interspersed throughout the manuscript to summarize mechanisms, highlight key concepts, and help readers navigate complex information. Having only one figure and one table in a review that spans ~16 pages makes the paper feel text-heavy and less engaging. It also limits the article’s value as a reference resource, since readers often rely on tables/figures for quick comprehension.
8. Figure 1 is mentioned (line 420), but from the truncated content it seems simplistic (just schematic arrows). It may not sufficiently synthesize complex pathways. A mechanistic diagram showing cross-talk between osteocytes, osteoclasts, osteoblasts, and PDL cells under force would add value.
9. Table 1 (lines 416–418) is helpful, but overly condensed. Some entries (e.g., LMHFV → NF-κB pathway) oversimplify results from cited studies.
10. The conclusion (lines 366–415) is too long and repetitive—essentially restating earlier sections. Future directions mention senolytics and genetic reprogramming (line 406) but lack discussion of realistic translational barriers (safety, regulatory hurdles, clinical feasibility).
1. Several typos and grammar issues are present (e.g., “visulaize” instead of visualize in line 35; “underappreciated” misspelled as underappreciated in line 364).
2. Inconsistent terminology (e.g., sometimes "OTM," sometimes spelled out "orthodontic tooth movement"). Needs uniformity.
3. The Abstract contains overly long sentences with multiple clauses (e.g., lines 31–38). This reduces readability and may confuse readers. Breaking them into shorter, precise statements would improve flow.
Author Response
Reviewer 3
Thank you for reading carefully and for the very helpful comments. I will answer them in order.
Comments and Suggestions for Authors
- While the review is comprehensive, it seems overly focused on TNF-α signaling and necroptosis, reflecting primarily the authors’ own recent publications. Other pathways (e.g., prostaglandins, nitric oxide, Wnt signaling) receive less attention, which skews balance.
ïƒ The focus of our review is on recent evidence regarding the role of osteocytes in OTM, particularly their contribution to osteoclastogenesis on the compression side. For this reason, we concentrated primarily on catabolic signaling pathways. Wnt signaling, being predominantly anabolic, falls outside the intended scope of this article. Nitric oxide and prostaglandins have been comprehensively covered in prior reviews (PMID: 34278439, PMID: 37792246, PMID: 37795174). Our intent is for this review to complement, rather than replace, those works by highlighting osteocyte-driven catabolic mechanisms in OTM.
- Some sections (e.g., on osteocyte morphology and mechano-adaptation, Section 5) are speculative with limited experimental evidence cited, yet the tone suggests strong conclusions.
ïƒ In the section on osteocyte morphology and mechanoadaptation, we intentionally draw a modest conclusion: “In conclusion, emerging evidence suggests that osteocyte morphology is a functional variable shaped by the mechanical environment, and its adaptation during OTM may be a previously underappreciated modulator of bone remodeling dynamics.”
The language throughout this section is deliberately measured. Rather than presenting definitive claims, we frame morphology as a possible, not established, modulator of remodeling. We explicitly ask questions such as “Yet the question remains: do osteocytes themselves structurally adapt in response to orthodontic force?” to underscore the exploratory nature of this section. Similarly, we consistently use phrasing like “These findings lend support to the hypothesis…” signals an interpretive rather than declarative tone. We use “could” over “does,” as in “such morphological plasticity could modulate site-specific remodeling efficiency and explain interindividual variability in treatment outcomes, particularly in aged or diseased bone.”
Taken together, we this our tone is restrained and appropriately reflective of the limited and emerging evidence base, and avoids strong conclusions.
- Multiple sections repeat the role of RANKL/OPG and TNF-α signaling (Sections 2 and 3, again in Section 6 conclusions). The narrative could be tightened by merging overlapping parts.
ïƒ Each section talks explicitly about a specific pathway, we acknowledge that RANKL/OPG pathway have been brought up in multiple sections but this highlights the multiple pathways through which RANKL/OPG signaling can be activated. If we combine it all in one section it will be overly complex. - The review highlights necrosis/necroptosis as an “alternative RANKL-independent mechanism” (lines 164–174) but does not adequately discuss limitations—most data are from mouse models, and human validation is absent.
ïƒ We do not mention necrosis in the review. We have reframed this section and removed the conclusion that necroptosis acts as an independent alternative RANKL pathway but actually works in synergy. - The LMHFV (low-magnitude high-frequency vibration) section presents vibration therapy as promising (lines 209–231), but the authors themselves admit limited evidence (rodents, short-term). Still, the conclusions read stronger than warranted.
ïƒ In lines 209-231 we discuss the evidence from the studies on LMHFV, and we do not draw any conclusions other than it is a promising therapy and we also point out the limitations, particularly the reliance on rodent models and short term studies: “The majority of studies are conducted in rodent models which may not fully recapitulate the complexity and mechanical environment of alveolar bone loading in humans, thereby limiting extrapolation to clinical orthodontics. The beneficial effects of vibration therapy (LMHFV) on accelerating OTM are promising and mechanistically clear but the results are largely based on small animal studies and short observation periods.”
We think this is a very balanced view and we would kindly ask the reviewer to indicate which specific statements they interpret as stronger conclusions than warranted. - Aging effects are presented as largely due to osteocyte senescence (lines 267–317), but alternative explanations (vascularity loss, systemic hormonal changes, reduced PDL turnover) are only briefly acknowledged.
ïƒ We thank the reviewer for pointing out to the evidence that other systemic factors are indeed crucial for ageing effects on bone remodeling not just in the context of orthodontic tooth movement but on bone metabolism in general. However this review as mentioned in the introduction tackles the recent evidence on osteocyte biology specifically in OTM specifically on the compression side in promoting osteoclast formation. Adding all the relevant ageing related factors will make the review extremely dense and very hard to read. - Table 1 and Figure 1 both appear only at the very end (in the Conclusions, Limitations, and Future Directions section). This is unusual for a review article, where figures and tables are normally interspersed throughout the manuscript to summarize mechanisms, highlight key concepts, and help readers navigate complex information. Having only one figure and one table in a review that spans ~16 pages makes the paper feel text-heavy and less engaging. It also limits the article’s value as a reference resource, since readers often rely on tables/figures for quick comprehension.
- Figure 1 is mentioned (line 420), but from the truncated content it seems simplistic (just schematic arrows). It may not sufficiently synthesize complex pathways. A mechanistic diagram showing cross-talk between osteocytes, osteoclasts, osteoblasts, and PDL cells under force would add value.
- Table 1 (lines 416–418) is helpful, but overly condensed. Some entries (e.g., LMHFV → NF-κB pathway) oversimplify results from cited studies.
ïƒ Response to 7 and 8: We added more tables and figures across the review to the body of the review (which less than 9 pages), to break up the text and improve readability.
ïƒ The table is a guide for readers to understand the jest of the recent evidence, and it is intended to be a simplified version of the text. For a more comprehensive and nuanced view on each subject the readers are invited to read the text in the body of the review. - The conclusion (lines 366–415) is too long and repetitive—essentially restating earlier sections. Future directions mention senolytics and genetic reprogramming (line 406) but lack discussion of realistic translational barriers (safety, regulatory hurdles, clinical feasibility).
ïƒ We discuss the future direction in light of translational barriers as the reviewer suggested.
Comments on the Quality of English Language
- Several typos and grammar issues are present (e.g., “visulaize” instead of visualize in line 35; “underappreciated” misspelled as underappreciated in line 364).
ïƒ The review has been scanned for grammar errors and typos. - Inconsistent terminology (e.g., sometimes "OTM," sometimes spelled out "orthodontic tooth movement"). Needs uniformity.
ïƒ We unified the style in the review. - The Abstract contains overly long sentences with multiple clauses (e.g., lines 31–38). This reduces readability and may confuse readers. Breaking them into shorter, precise statements would improve flow.
ïƒ The abstract was rewritten and shortened.
Round 2
Reviewer 3 Report
Comments and Suggestions for Authors
The revision successfully addresses the majority of my concerns. The manuscript is now more balanced, clearly written, and significantly improved in structure and readability. Minor residual issues include some redundancy in pathway discussion and a still limited integration of systemic aging factors. However, these do not detract substantially from the manuscript’s quality.
Author Response
Reviewer 3
The revision successfully addresses the majority of my concerns. The manuscript is now more balanced, clearly written, and significantly improved in structure and readability. Minor residual issues include some redundancy in pathway discussion and a still limited integration of systemic aging factors. However, these do not detract substantially from the manuscript’s quality.
- We sincerely thank the reviewer for their considerable contribution to bettering the manuscript. The recurrence of pathway discussion is deliberate, as signaling mechanisms in osteocyte-driven osteoclastogenesis are inherently complex. Reiterating them across sections is our way of improving readability and guiding the reader through this complexity.
- As for systemic aging factors, addressing them comprehensively would make the review unnecessarily dense. Since the aim of this work is to highlight the most recent evidence in the specific niche of osteocyte–osteoclastogenesis in OTM, we view this as beyond the scope of the present manuscript. The broader effects of aging are, however, acknowledged in lines 322–327.